# Secretome and Tunneling Nanotubes: A Multilevel Network for Long Range Intercellular Communication between Endothelial Cells and Distant Cells

**DOI:** 10.3390/ijms22157971

**Published:** 2021-07-26

**Authors:** Béatrice Charreau

**Affiliations:** Inserm, Centre de Recherche en Transplantation et en Immunologie, Université de Nantes, UMR 1064, ITUN, F-44000 Nantes, France; Beatrice.Charreau@univ-nantes.fr

**Keywords:** endothelial cells, intercellular communication, tunneling nanotubes, exosomes, extracellular vesicles, secretome, cytokines, chemokines, ectodomain shedding, secretory pathway

## Abstract

As a cellular interface between the blood and tissues, the endothelial cell (EC) monolayer is involved in the control of key functions including vascular tone, permeability and homeostasis, leucocyte trafficking and hemostasis. EC regulatory functions require long-distance communications between ECs, circulating hematopoietic cells and other vascular cells for efficient adjusting thrombosis, angiogenesis, inflammation, infection and immunity. This intercellular crosstalk operates through the extracellular space and is orchestrated in part by the secretory pathway and the exocytosis of Weibel Palade Bodies (WPBs), secretory granules and extracellular vesicles (EVs). WPBs and secretory granules allow both immediate release and regulated exocytosis of messengers such as cytokines, chemokines, extracellular membrane proteins, coagulation or growth factors. The ectodomain shedding of transmembrane protein further provide the release of both receptor and ligands with key regulatory activities on target cells. Thin tubular membranous channels termed tunneling nanotubes (TNTs) may also connect EC with distant cells. EVs, in particular exosomes, and TNTs may contain and transfer different biomolecules (e.g., signaling mediators, proteins, lipids, and microRNAs) or pathogens and have emerged as a major triggers of horizontal intercellular transfer of information.

## 1. Introduction

Lining all blood vessels of the vascular tree, the endothelium represents a surface area of around 5000 m^2^ composed of roughly 10^14^ endothelial cells (ECs) and can be perceived as an organ system [1]. At the interface between the blood and tissues, the EC layer is involved in the control of key functions including vascular tone, permeability and homeostasis, leucocyte trafficking and hemostasis [2,3]. To ensure this broad variety of functions in different vascular beds and tissues, ECs display marked heterogeneity in both structure and function [4,5,6,7,8,9]. At rest, in physiological conditions, quiescent ECs exhibit specific phenotype and functions. Quiescence is not an inactive state, but instead corresponds to an active state [10,11]. The maintenance of functionally quiescent EC monolayer is under the control of multiple integrated signaling pathways [12]. As a gatekeeper, EC respond to a large variety of stimuli by adapting both their phenotype and their functions to preserve vascular integrity [13,14,15]. To this aim, EC display at the cell surface specific receptors that will sense extracellular signals to initiate specific changes in EC phenotype and functions. Usually, these changes are transient and EC return to a quiescent state after stimulation. ECs receive numerous signals ranging from mechanical to cell–cell contact to paracrine and endocrine inputs [1,2].

Intercellular communication is a fundamental process in the development and functioning of multicellular organisms and to operate effective mobilization and recruitment of circulating cells at sites of inflammation, infection, or injury. An intensive crosstalk between ECs and circulating hematopoietic cells and other vascular cells is needed to maintain vessel structure, function and homeostasis. Major mechanisms governing EC interaction with adjacent cells in tissue or blood vessels include exocytosis and direct transfer of small cytoplasmic components via gap junction [16,17]. For communication with more distant cells, ECs must use additive mechanisms to counteract the diluting effect mediated by the constant flow of blood. To achieve long-distance communications, several strategies can be used, including the secretion of messengers such as cytokines, chemokines, complement proteins, coagulation or growth factors, the release of extracellular vesicles (EVs), or the formation of thin tubular membranous channels termed tunneling nanotubes (TNTs), which may contain and transfer different biomolecules (e.g., signaling mediators, proteins, lipids, and RNAs) or pathogens.

The aim of this review is to provide an overview of some of the advances in our molecular and functional understanding of the secretory pathways, granules and extracellular vesicles, generating the endothelial secretome, and of TNTs used for intercellular communication between ECs and distant cells.

## 2. Endothelial Exocytosis

For communication ECs transport a large number of proteins to the plasma membrane and the extracellular medium by using the secretory pathway. The secretory pathway is the place where synthesis and delivery of soluble proteins occur. In ECs, the protein secretory pathway is an essential molecular machinery to prepare and export proteins into the extracellular environment. The secretory pathway includes several functional modules that are compartmentalized along the ER and Golgi apparatus. The secretory pathway comprises the rough endoplasmic reticulum (rough ER), ER exit sites (ERESs), the ER-to-Golgi intermediate compartment (ERGIC), the Golgi complex and post-Golgi carriers. The organelles of the secretory pathway have a precise organization and structure in order to sustain their function in folding, processing of the post-translational modifications (PTMs), and trafficking of the proteins to the membrane of extracellular space [18,19].

### 2.1. Secretory Pathway and Weibel–Palade Bodies

To provide a rapid answer to external stimulation or stress, ECs are equipped with specific rod-shaped organelles known as Weibel–Palade bodies (WPBs) in which several bioactive molecules are stored and rapidly released upon stimulation (Figure 1). The first description of WPBs was provided by E. R. Weibel and G. E. Palade who described the dimension (diameter of 0.1–0.3 μm, length of 1–5 μm) and the tubular nature of this organelle using transmission electron microscopy [20]. WPBs are the best characterized secretory organelles of ECs [21].

The most abundant cargo in WPBs is von Willebrand factor (VWF). VWF is a multimeric glycoprotein, which mediate platelet adhesion and aggregation at sites of endothelial damage and is a carrier for coagulation factor VIII and to the subendothelial matrix [22,23]. Plasma levels of VWF constitute a key regulator of thrombosis, and elevated levels of VWF are associated with an increased risk of thrombosis. The release of VWF can either be apical or luminal into the circulating blood, basolateral or abluminal into the subendothelial matrix. Under conditions of low shear stress, platelet adhesion to the damaged vessel wall is mediated by ultra-large VWF multimers released from ECs, which form extremely long string-like structures that tether platelets [24]. However, under conditions of high shear stress, platelet aggregation occurs only in the presence of soluble VWF. VWF is not only a cargo, but also a prerequisite for the formation of WPBs. ECs of VWF-deficient mice do not contain WPBs [25].

After their formation at the trans-Golgi network (TGN), WPBs accumulate in the cytoplasm and can remain within ECs for long periods of time (few days) [26]. These features of WPBs are consistent with a function as a storage organelle. Under resting conditions, WPBs undergo a very slow process of basal exocytosis. However, their rate of exocytosis is rapidly increased in response to external stimuli that elevate intracellular free calcium (Ca^2+^) concentrations [27]. In ECs, the vesicle-associated membrane protein (VAMP)-3 cooperates with synaptosomal-associated protein 23 (SNAP23) to direct the fusion of Weibel–Palade bodies with the plasma membrane, and thus mediates the exocytosis of VWF [28].

Several proteins have been found copackaged with VWF including P-selectin, Rab27a, endothelin 1, angiopoietin-2, or CD63 [29,30]. P-selectin has been found to anchor newly released ultra-large VWF to the surface of activated ECs and to present the VWF cleavage sites to ADAMTS-13 [31]. Local EC microenvironment triggers a dynamic control over the content of WPBs by selectively including or excluding certain cargoes. The content of WPBs is modulated by shear stress and by inflammation [32]. In addition to VWF, several inflammatory and angiogenic factors have been found copackaged into WPBs [33,34]. The storage of EC-derived chemokines that may attract other subsets of leukocytes has been reported upon inflammation including CXCL1, CCL2, CCL5, CCL26 and CXCL10 and CCL5 [33]. However, the relevance of chemokine storage in WPBs has recently been questioned. The efficiency of sorting to WPBs for tPA, IL-8, CCL2, and CXCL1 was found weak suggesting an inefficient storage for these chemokines which may be accidentally included into WPBs and inefficiently removed due to their ability to bind to VWF at low pH as observed for IL-8 [21]. It may also reflect the heterogeneity of WPB content according to the EC activation state at the time of WPB synthesis [30]. Originally being identified in arterial ECs, it has been proposed that both the distribution and the content of WPBs may vary according to the localization of ECs within the vascular tree. Consistent with this hypothesis, elevated numbers of VWF were found in the pulmonary artery [35]. In view of these findings, it is anticipated that the content of WPB will reflect EC heterogeneity. However, a comparative analysis of WPBs across ECs from different vascular beds and tissue remains to be established.

### 2.2. Endothelial Exocytosis of Secretory Granules

Differential release of various granule populations is a well-known feature of certain regulated secretory cells and it has been described for ECs. Stimulus-release coupling occurs via regulated exocytosis of secretory vesicles and Ca^2+^ was a key trigger of this process [36]. The N-ethylmaleimide-sensitive factor (NSF) protein is a key player in the multiple steps of both the constitutive secretory and endocytic pathways acting in ER to Golgi transport and in endocytic vesicle fusion [37]. Cytokines, chemokines, enzymes, growth factors, matrix proteins, and signaling molecules can all be secreted by fusion of the secretory granule with the plasma membrane and the subsequent release of the vesicular contents (Figure 1) [36]. Previous studies have identified proteins that mediate the regulated exocytosis in ECs: vesicle-associated membrane protein (VAMP) 3, VAMP-8, syntaxin 4, syntaxin-binding protein 5, and synaptosomal-associated protein 23 (SNAP23) [28,38,39]. Most of these proteins belong to the superfamily of transmembrane proteins named soluble N-ethylmaleimide attachment protein receptors (SNAREs) that mediate trafficking of cellular contents between intracellular compartments. Membrane docking and fusion require one SNARE molecule on a granule membrane bound to two counterpart SNAREs on a target membrane, forming a stable ternary complex that mediates granule exocytosis [40].

Cytokines and other soluble mediators can be involved in distant as well as local communication. In humans, about 2600 genes—corresponding to approximately 13% of all protein-coding genes—code for potentially secreted proteins, and around 500 of these proteins were predicted to be secreted in the proximity to the cell of origin. ECs secrete a range of paracrine, autocrine, and endocrine factors that mediate multiple aspects of EC function. An intensive crosstalk between cells via soluble factors, including cytokines, contributes to vessel structure, function, and maintenance, proliferation of ECs and vascular smooth muscle cells, EC-leukocyte interactions, platelet adhesion, coagulation, inflammation, and permeability [41]. Consequently, the endothelium is a major secretory organ, releasing proteins both into the circulation and also the vascular matrix. A proteomic analysis showed that ECs polarize secretion of vascular matrix components to the basolateral surface while the apical EC proteome is primarily composed of proteins secreted through the extracellular vesicle pathway [42].

Regulated exocytosis provides a means by which ECs can very rapidly and selectively alter the local vascular microenvironment and modulate the interconnected processes of coagulation, fibrinolysis, and inflammation. Endothelial exocytosis of granules is a rapid response to vascular injury. During exocytosis, vesicles deliver proteins from the Golgi to the extracellular space. The proteins that control vesicle and granule trafficking in ECs have been only partially identified. Syntaxin 4, synaptobrevin 3, and NSF play a critical role in endothelial granule exocytosis [43]. Additional studies have shown that nitric oxide (NO) also regulates exocytosis by chemically modifying NSF. Concerning the regulation of exocytosis, Zhu and colleagues demonstrated that the syntaxin-binding protein STXBP5 inhibits exocytosis in ECs. STXBP5 interacts with syntaxin-4, a component of SNARE complexes, and synaptotagmin, the sensor for mediating calcium-induced exocytosis [38]. Polarized secretion is controlled by the cytoskeletal linker protein liprin-α1 [42].

ECs continuously communicate with other cells through a variety of mechanisms. A key mode of communication is the release of soluble cytokines and chemokines. These signals lead to the induction of biological processes, such as cell proliferation, differentiation, and migration that in turn influence the homeostasis of the tissues and organs by controlling the cell types and numbers that are produced [44]. The extracellular propagation of the signal is mediated by random molecular motion of the soluble cytokines- and chemokines. Although the cells cannot control diffusion, they can regulate their biological processes such as protein production and secretion rates. A single cell model was used by K. Francis and B.O. Palsson to estimate the effective communication distance over which a single circulating cell can meaningfully propagate a cyto/chemokine-mediated signal [45]. The analysis revealed that the domain within which a unique cell can effectively communicate is ≈250 μm in size; and the communication within this domain takes place in 10–30 min. Interestingly, it was determined that an adherent cell such as ECs secrete extracellular proteins into a hemispherical fluid domain. The effective communication distance is drastically increased compared to non-adherent cells and is dependent on its location in tissue or blood vessels. There are several published experimental observations that are consistent with this communication distance estimates. A time-lapse study of the motility characteristics of bovine pulmonary artery ECs showed that cells appear to sense the presence of one another at distances of ≈40 μm, perhaps by communicating using soluble cytokines or chemokines [46]. Furthermore, it has been shown that hematopoietic progenitor cells (CD34^+^ cells) can sense each other over distances of ≈100 μm [47].

ECs have been shown to express interleukin-1 (IL-1), IL-5, IL-6, IL-8, IL-11, IL-15, several colony-stimulating factors (CSF), granulocyte-CSF (G-CSF), macrophage CSF (M-CSF) and granulocyte-macrophage CSF (GM-CSF). Secretion of chemokines that may attract selectively subsets of leukocytes have been reported upon inflammation [15,44]. This set of chemokines includes CXCL1 (attracting neutrophils), CCL2 (monocytes), eotaxin-3/CCL26 (eosinophils), CXCL10 (T cells), and regulated on activation, CCL5 (eosinophils, monocytes, and T cells) and regulated on activation, CCL5 (eosinophils, monocytes, and T cells) [33]. Chemokines belong to a large family of small (8 to 12 kDa) cytokines that direct leukocyte migration during development, homeostasis, and disease [44,48]. There are 4 chemokine subfamilies (C, CC, CXC, and CX3C) distinguished and named by the number and arrangement of conserved cysteines near the N terminus [44]. In the classic multistep model of leukocyte transendothelial migration, chemokines act by binding sequentially to 2 main classes of molecules: glycosaminoglycans (GAGs) on ECs and 7-transmembrane domain G protein-coupled receptors (GPCRs) on leukocytes. GAGs act as extracellular scaffolds that concentrate chemokines on the luminal surface of post-capillary venules, positioning them to attract circulating leukocytes by activating specific leukocyte GPCRs [49]. GAGs contain highly anionic sulfated polysaccharides and are thought to protect chemokines from proteases and shear forces in blood vessels. Of note, GAG binding promote the formation of chemokine oligomers with a higher GAG-binding affinity than monomers [50]. These processes increase presentation of endothelial chemokines to leukocytes.

Upon injury, ECs contribute to the development of fibrosis through the release of profibrotic mediators, such as TGFβ, plasminogen activator inhibitor 1 (PAI-1) and connective tissue growth factor, which induce fibroblast growth and differentiation and collagen synthesis by fibroblasts [51]. ECs experiencing disturbed flow can deliver miRNAs, such as miR-126-3p and miR-200a-3p, to recipient cells such as smooth muscle cells (SMCs) in forms independent of membrane-bound vesicles [52]. Many miRNAs are released from ECs in complexes with argonaute-2 (AGO2), which protect miRNAs from ribonuclease digestion and facilitate their uptake by the recipient cells [53].

## 3. Endothelial Cells and Extracellular Vesicles

### 3.1. Definition and Biogenesis of EVs

Intercellular connection by extracellular vesicles (EVs) is another way of cell to cell crosstalk that allows cells to deliver biological contents and messages to distant recipient cells. ECs constitutively secrete EVs into the blood in low concentrations under physiological conditions. It has been suggested that endothelial EVs account for approximately 5–15%, and constitute a large subclass of all circulating EVs in blood, albeit the majority of circulating plasma EVs are derived from platelets and erythrocytes [54]. However, the amount of EVs released by ECs remains difficult to estimate due in part to sampling artifacts and to a lack of standardization of the techniques for the isolation and characterization of EVs [55]. Plasma levels of endothelial EVs have been found to be elevated in various diseases involving EC injury or dysfunction [56].

The generic term ‘extracellular vesicles’ (EVs) was proposed by György, B. et al. in 2011 [57] to define all lipid bilayer-enclosed extracellular structures. EVs can form by external budding of the plasma membrane or by an intracellular endocytic trafficking pathway involving fusion of multivesicular late endocytic compartments (multivesicular bodies, MVBs) with the plasma membrane. EVs are composed of a lipid bilayer containing transmembrane proteins and enclosing cytosolic proteins and RNA. Cells can secrete different types of EVs that have been classified according to their sub-cellular origin [58].

Nevertheless, specific tools to distinguish EVs of different intracellular origins, and thus probably different functions are still lacking. Consequently, EV classification, formation mechanisms and contents are still evolving. In the present review we will focuses on two major types of EVs: microvesicles (MVs) and exosomes (Figure 1).

MVs are large vesicles (100–1000 nm) that may display a diverse range of sizes and are released from the plasma membrane by budding. In contrast, exosomes are smaller vesicles (50–150 nm) that originate from the endosome [59]. MVs include a relatively heterogeneous population of vesicles. MV formation is regulated by membrane lipid microdomains and regulatory proteins such as ADP-ribosylation factor 6 (ARF6) [60]. Exosomes constitute a homogeneous population of small vesicles that are formed by inward budding of the multivesicular body (MVB) membrane. Exosome synthesis involves mainly the endosomal sorting complex required for transport (ESCRT) protein, lipid ceramide and neutral sphingomyelinase [61]. Cargo sorting into exosomes involves ESCRT and associated proteins including tumor susceptibility gene 101 protein (TSG101), ALG-2-interacting protein X (Alix) and the small GTPases such as Rab7a and Rab27b [62]. Exosomes are released into the extracellular space after fusion of MVBs with the cell membrane, a process regulated by the GTPases Rab27Aa, Rab11, and Rab31 [62].

All EVs bear surface molecules that allow them to be targeted to recipient cells. Once attached to a target cell, EVs can induce signaling via receptor–ligand interaction or can be internalized by endocytosis and/or phagocytosis or even fuse with the target cell’s membrane to deliver their content into its cytosol, thereby modifying the physiological state of the recipient cell [59,63]. An integrin-associated protein, CD47, which protects cells from phagocytosis, is often found on the surface of EVs and was shown to increase the time of EV circulation in the blood by preventing their phagocytosis by macrophages and monocytes [64].

Exosomes and microvesicles are secreted by all cell types and found in most body fluids. The content of EVs usually reflects the status of the donor cell and can influence the behavior of recipient cells both locally and systemically [63,65]. In addition to ECs, most circulating cells of the vascular system, such as red blood cells, platelets, monocytes, lymphocytes, dendritic cells, and mast cells, secrete exosomes under normal conditions. Under pathologic or stress conditions, exosome secretion may increase and/or exosomal content may change [66]. Exosomes are characterized and identified by their expression of a set of integrins and tetraspanins (CD9, CD63, CD81, and CD82), proteins involved in membrane transport and fusion (annexins, Rab proteins, and flotillin), proteins associated with multivesicular body biogenesis (Alix, TSG101), and heat shock protein (HSP)-70 and -90 [67].

However, it should be noted that while these are hypothetical mechanisms, in actual samples that are harvested from different biological sources, a population of small particles can be found that is heterogeneous in size, shape and composition, and when already present in the isolate, the origin of the particles is usually not evident. Formation of EVs due to processing of samples is often overlooked, although it may present a considerably portion of EVs in the samples.

### 3.2. Mechanims of EV Uptake and Cargo Delivery

Similar to EV formation, the mechanisms of EV uptake and cargo delivery into the cytosol of recipient cells also remain incompletely elucidated. It has been proposed that the three steps of EV uptake involves (1) the targeting of the recipient cell, (2) the internalization of EVs by recipient cells and (3) the delivery of EV content to the recipient cell. Whether EV targeting recipient cells is a cell specific mechanism or, in contrast, results from a generic targeting process is still debated. For instance, EVs from oligodendrocytes are internalized preferentially by microglia, rather than neurons [68], while in contrast, the HeLa epithelial cells are able to take up EVs produced by different cells [69]. Similarly, whether internalization of EVs by recipient cells occurs through a specific, receptor-dependent pathway, or through a non-specific process such as micropinocytosis or macropinocytosis is unclear. Several molecules localized on the surface of both EVs and recipient cells have been proposed to be involved in cell uptake and to contribute to some degrees to its specificity. They include integrins, lectins/proteoglycans and the T cell immunoglobulin and mucin domain-containing protein 4 (Tim4) [70,71,72]. EV internalization has been reported to occur through multiple routes that involve endocytosis [63]. Concerning the cargo delivery mechanisms, the endosome is the putative location of EV-content delivery, and membrane fusion in response to acidic pH has been proposed as a possible mechanism similarly to the process used by some viruses [73,74]. Other mechanisms such as direct transfer into the nucleus or the ER through the contact of these compartments with endosomes that contain internalized EVs have been also reported.

In ECs, EV uptake has been shown to be mediated via the interaction of EV surface proteins such as tetraspanins with membrane receptors of the recipient cells [75]. Tumor-derived EVs bearing Tspan8-CD49d complexes have been shown to be internalized by vascular rat ECs, thereby enhancing EC migration, proliferation and sprouting [76]. EVs bearing Tspan8-α4 complexes were also found incorporated by rat ECs after binding to intercellular adhesion molecule (ICAM)-1 [77]. EVs can also transmit information to recipient cells by EV/cell surface contact, without delivery of their content. As an example, during immune responses, EVs harboring major histocompatibility complex (MHC)–peptide complexes can activate cognate T cell receptors on T lymphocytes [78].

### 3.3. Endothelial Cell-Derived EVs

ECs constitutively secrete EVs into the blood. Under physiologic conditions, ECs are subjected to laminar shear stress (SS), which is required for maintaining normal vascular function by exerting anti-coagulant, anti-inflammatory, and vasodilatory effects through the release of NO [79]. Some studies reported a strong correlation between changes in laminar SS and the release of endothelial MVs levels in vivo with elevated levels of endothelial EVs due to decreased SS and vice et versa [80,81,82].

EC injury triggers the release of EC-derived EVs including both exosomes and MVs. Upon inflammation, the proinflammatory cytokine TNF leads to a dose-dependent increase in the release of endothelial EVs, which could be reversed by the use of blocking anti-TNF antibody [83]. Moreover, TNF promotes an induction of tissue factor (TF) on the surface of the endothelial EVs [83]. Stimulation of ECs with other inflammatory factors, including interleukin-1 (IL-1), interferon-γ (IFN-γ) and bacterial lipopolysaccharide have also been shown to induce the release of endothelial EVs containing higher levels of specific miRNAs than endothelial EVs derived from unstimulated ECs [84]. In addition to proinflammatory cytokines, coagulation factors such as thrombin [85], C-reactive protein (CRP) [86], and plasminogen activator inhibitor-1 (PAI-1) [87] are also capable of inducing the release of EVs from ECs [56]. Hypoxia and LPS have furthermore been shown to increase the release of exosomes from pulmonary artery ECs, promoting proliferation and resistance to apoptosis in pulmonary artery smooth muscle cells [88]. In addition, ECs stimulated with transforming growth factor (TGF)-β induced the shedding of VEGFR2-containing exosomes, which regulate the effects of angiogenic stimuli on vascular sprouting [89].

### 3.4. EVs Trigger Endothelial Protection and Vascular Repair

Several studies have shown that EVs of endothelial origin can act to promote EC and vascular integrity in vitro and in vivo. Abid Hussein et al. have demonstrated that endothelial MV release is cell protective by exporting caspase-3, one of the effector enzymes of apoptosis, into MVs and thereby diminishing intracellular levels of proapoptotic caspase-3 [90]. In a similar way, Jansen et al. have demonstrated that incorporation of annexin I/phosphatidylserine receptor-dependent endothelial MVs by ECs protects ECs against apoptosis by inhibiting MAPkinase p38 activity [91]. MVs derived from platelets also positively influence endothelial regeneration by at least two non-exclusive mechanisms. They could directly interact with ECs and promote vascular regeneration, or they may activate endothelial progenitor cells (EPCs) to facilitate endothelial repair and vascular regeneration [91,92]. Endothelial MVs carrying the coagulation factors endothelial protein C receptor (EPCR) and activated protein C (APC) display cytoprotective and anti-inflammatory effects [93]. Thus, EC-derived EVs can be viewed not only as markers of vascular integrity, but also as relevant effectors in intercellular vascular signaling. EC-derived EVs can contribute to intercellular communication during the development of atherosclerosis via the transfer of cellular contents such as protein and miRNA, which may prevent or promote disease progression depending on the context [94].

### 3.5. Transfert of miRNAs via EVs

EVs are fully active membrane vesicles that contain and transfer functional intracellular contents. EVs convey cellular messages through their distinct cargoes consisting of proteins, cytokines, mRNA, or noncoding RNA such as microRNA (miRNA) or long noncoding RNA (lnc RNA) to target cells and influence their function and their phenotype [95]. Among the biological contents transferred by EVs into target cells, miRNAs play a key role by controlling mRNA and protein expression in recipient cells [54]. MiRNAs regulate the expression of mRNAs at post transcriptional level either via translational repression or mRNA degradation [96]. It has been speculated that exosome-coupled miRNAs are not passively released from the cells but, in contrast, selectively may be loaded into exosomes to play specific functions. Numerous in vitro studies have demonstrated that extracellular miRNAs enwrapped with exosomes can alter gene expression in the recipient cells such as ECs. For instance, exosomes derived from hypoxic leukemia cells contain a subset of upregulated miRNAs including miR-210 which may enhance tube formation in ECs [97]. Like exosomes, MVs can also carry miRNAs. The presence of miRNAs in MVs has been reported from different cells including ECs [98].

Delivery of functional miRNA-126 into recipient ECs mediated by endothelial MVs improves vascular endothelial repair [99]. Exosomes released from cardiac progenitor cells increase the migratory capacity of ECs in vitro [100]. EVs released from endometrium-derived mesenchymal stem cells transfer miR-21 into ECs, thereby exerting cardioprotective and proangiogenic effects [101]. ECs have also been shown to internalize miRNA-enriched EVs derived from monocytes/macrophages, which cause regulation of target gene expression and EC function, as well as enhanced EC migration [102,103]. Furthermore, internalization of hepatocyte-derived EVs into ECs induces endothelial dysfunction, resulting from arginase-activity provided by EVs [104]. Concerning miRNA, Rab GTPases have been shown as central coordinators of membrane trafficking with distinct members of this family being responsible for specific transport pathways. A vesicular export mechanism for miR-143, induced by the shear stress responsive transcription factor KLF2 was identified and is dependent on Rab7a/Rab27b in ECs [62].

Endothelial exosomes are capable of transferring miRNAs such as miR-503 to tumor cells, which decreased tumor cell proliferation and invasion in vitro [105]. ECs secrete exosomes containing the Notch ligand Delta-like 4, thereby promoting angiogenesis via inhibition of Notch signaling [106]. High glucose culture of glomerular ECs led to increased levels of exosomes, and also activated glomerular mesangial cells and promoted diabetic nephropathy via transfer of TGF-β1 mRNA [107]. Increased exosome secretion by senescent human ECs has been shown to impair osteogenesis of human mesenchymal stem cells (MSCs) in vitro by transfer of its selective cargo while miR-31 is overrepresented in senescent EC-derived exosomes and inhibitory to osteogenic differentiation [108], the osteogenesis-promoting protein galectin-3 is underrepresented in EC-derived exosomes [109]. This suggests EC-derived EVs also cross-talk within the bone marrow niche and are involved in the pathogenesis of osteoporosis, as circulating miR-31 is also found to be higher in individuals with osteoporosis [108].

### 3.6. EVs and Immune Responses

Evidence is emerging that extracellular vesicles can act as triggers and regulators of immune responses in particular in cancer [110]. Indeed, besides molecules commonly found in most exosomes (adhesion molecules, heat shock proteins, molecules involved in membrane trafficking and ESCRT, etc.), molecules with specialized function in immune responses like MHC class I and II, costimulatory molecules and several immune receptors and ligands (such as FasL, TRAIL, PD-L1, NKG2D ligands) have been identified in the cargo of various exosomes. ECs express not only MHC class I and class II and most of costimulatory molecules displayed on antigen presenting cells such as dendritic cells [111], but also a set of non-classical and MHC class-I–like molecules including HLA-E, MICA and other NKG2D ligands that may potentially activate cognate receptors on natural killer (NK) and T cells [112,113]. EVs mediated intercellular communication between monocytes and ECs might play a major role in vascular inflammation and atherosclerotic plaque formation during cardiovascular diseases. EVs derived from TNF-induced activated vascular ECs contain a pro-inflammatory profile with chemotactic mediators, including intercellular adhesion molecule (ICAM)-1, CCL-2, IL-6, IL-8, CXCL-10, CCL-5, and TNF-α, and were readily taken up by recipient monocytes THP-1 and HUVEC. EVs released from inflamed ECs are able to establish a targeted cross-talk between EC and monocytes and reprogramming them toward a pro- or anti-inflammatory phenotypes [114]. Endothelial MVs also display anti-inflammatory effects in vitro and in vivo by reducing endothelial ICAM-1 expression by the transfer of functional miRNA-222 into recipient cells [101]. ECs suppress monocyte activation through secretion of extracellular vesicles containing anti-inflammatory microRNAs. In particular, miR-10a was transferred to monocytic cells from EC-EVs and could repress inflammatory signaling through the targeting of several components of the NF-kB pathway, including IRAK4 [115].

LPS induces neutrophils to secrete exosomes containing miR-122-5p with oxidative stress, apoptosis and increased permeability of brain microvascular ECs [116]. Hardy et al. reported that EVs released by apoptotic ECs have a distinct transcriptomic profile and carry non-coding RNA sequences exhibiting immunostimulatory potential, including mitochondrial transfer RNAs, U1 small nuclear RNA, and pathogen-like endogenous retroelements. Apoptotic ECs release EVs loaded with RNAs, which are recognized by RIG-I-like receptors and endosomal Toll like receptors (TLR3, TLR7 and TLR8), and therefore have the ability to elicit innate immune responses [117].

## 4. Shedding of Endothelial Protein Ectodomains

In addition to the proteins that undergo protein secretion via secretory pathways, several membrane proteins are known to be released into the extracellular space via ectodomain shedding [118]. Previous studies on membrane proteins revealed that about 2–4% of cell surface molecules undergo the shedding process [119,120]. Several membrane-bound EC adhesion molecules, growth factors, cytokines, and cell receptors can be proteolytically cleaved by sheddases that results in the release of soluble forms of fragments. The process of ectodomain shedding has been shown to regulate vascular pathologies and diseases such as degeneration, inflammation, cancer and physiological processes such as proliferation, differentiation, and migration.

A recent database for the identification of shed membrane proteins revealed that a largest portion of the shed membrane proteins (34%) were found to be related to immune response, blood, homoeostasis and angiogenesis consistent with a major contribution of shedding in vascular biology [121]. Proteolytic removal of membrane protein ectodomains (ectodomain shedding) is a post-translational modification (PTM) that controls levels and function of numerous membrane proteins. The contributing proteases, referred to as sheddases, act as important molecular switches in processes ranging from signaling to cell adhesion. The shedding process consists on the proteolytic removal of membrane protein ectodomains (ectodomain shedding) by a protease (referred to as sheddase) which cleaves a membrane protein substrate close to or within its transmembrane (TM) domain. This results in release of the soluble ectodomain from the membrane and a fragment that remain bound to the membrane.

Proteases are commonly considered as «canonical» sheddases if they cleave their substrates in the luminal juxtamembrane domain with a short distance to the membrane-anchoring domain [122] while «non-canonical» sheddases cleave within a TM domain or at the membrane boundary. The best-characterized sheddases are expressed by ECs and they include two members of the “a disintegrin and metalloprotease” (ADAM) family ADAM10 and, ADAM17 (also called TACE for TNFα-converting enzyme), and “β-site APP cleaving enzyme” (BACE1) [123]. In addition ECs also express several matrix metalloproteases (MMPs), which cleave soluble proteins or remove pro-peptides without shedding the whole ectodomain. ADAM10 and ADAM17 are most likely active in the trans-Golgi network, in the later secretory pathway compartments, and at the plasma membrane. Substrate cleavage by ADAM10 often occurs in basal conditions and does not require cell stimulation, whereas shedding by ADAM17 is mostly observed when ECs are stimulated [124]. ADAM10 is implicated in proteolytic events and in the cleavage of various substrates on ECs. Among them CD44, Interleukin 6-receptor (IL-6R), CX3CL1 (fractalkine), CXCL16, VEGFR2, DLL4 and VE-cadherin are of special importance for vascular biology [125,126]. ADAM10 was shown to cleave VCAM-1 on TNF-activated ECs [126]. Upon inflammation, ADAM10 and ADAM17 have been shown to promote cellular contact between inflammatory cells and ECs by reducing the cell-surface level of endomucin, a component of the glycocalyx layer, which interferes with the interactions between these cells [127].

The consequences of shedding events for vascular cells strongly depend on the type of shed molecule resulting in the generation of soluble receptor agonists and antagonists [128]. For example; ADAM10 was shown to cleave the extracellular domain of ICAM-1 from the endothelial surface, allowing neutrophils to cross the endothelium during the final diapedesis step [129]. Interestingly, the soluble forms of these adhesion molecules display a dominant negative effect inhibiting the interaction with the membrane-bound forms of these adhesion molecules [129]. ECs contribute to both innate adaptive immune responses by expressing classical MHC antigens and costimulatory molecules [111], but also non-classical l MHC class I, such as the MHC class I related chain A (MICA) and HLA-E molecules [112,130,131]. ADAM10 cleave MICA molecules from tumor cell and EC surface generating a soluble NKG2D antagonist impairing activation of NK and T cells, and thereby inhibiting immune responses [131,132]. Previous studies have suggested that the relative distribution of circulating forms of soluble MICA between exosomes and shedded forms may have prognostic value. Thus, ectodomain shedding probably serves important regulatory events at EC and vascular level besides transcription regulations.

## 5. Endothelial Cells and Tunneling Nanotubes

For long distance intercellular communications, ECs can also form «tunneling nanotubes» [133]. TNTs are thin (originally reported as ranging from 50 to 200 nm) membranous nanotubes, containing F-actin enclosed in a lipid bilayer, and able to transfer molecular information by forming a “tunnel” between distant cells (Figure 2 and Figure 3) [134]. TNTs have been initially reported in Drosophilia [135], and before described in a rat cell line (PC12 cells) in 2004 by Rustom and colleagues [134]. TNTs have been observed in multiple cell types in vitro [136] and in vivo [137]. In contrast to soluble factors or microvesicles that diffuse and decrease over distance, TNTs propagate signals through a network of cells that remain strong and robust despite the distance traveled [138,139]. TNT permit the direct exchange of various components or signals (e.g., ions, proteins and organelles) between non-adjacent cells at distances over 100 μm. TNTs allow connected cells to act in a synchronized manner over long distances, with some interactions on the scale of hundreds of microns away [140,141].

### 5.1. TNT Structure and Biogenesis

TNTs are heterogeneous in terms of diameter, length, and cytoskeletal composition [133,142,143,144]. TNTs are broadly defined as actin-containing, non-adherent, intercellular membranous connections that allow intercellular transfer of molecular information. All TNTs contain filamentous F-actin as backbone, but TNTs with the smaller diameter (<0.7 μm, “thin” TNTs) only contain F-actin, whereas “thick” TNTs (>0.7 μm) often contain both actin and microtubules [145].

At least two models of TNTs formation have been proposed. Firstly, TNTs can result from a filopodium-like protrusion that elongates and docks on a neighboring cell and further differentiates into a TNT [134]. In another model, TNTs originate between two connected cells that move away from each other, but retain a thin membranous thread [145,146]. TNTs are usually open at both ends they display no adhesion properties and use distinct actin regulators for their formation, which are some of the properties that distinguish them from other similar cellular extensions like filopodia.

The duration of the intercellular communication has not been yet studied in ECS but has been recently reported for cancer cells [147]. The range of average duration of TNTs was shown to be of 30 min to 2 h, but the majority (>80%) of TNTs lasted no longer than 30 min and none lasted longer than 2 h. Interestingly, the overall proportion of cells that developed intercellular interactions with TNTs was small (7%) within the first 10 h in culture.

### 5.2. Mechanisms of TNT Formation

Molecular mechanisms that underlie the formation and function of TNTs are still poorly understood (reviewed in [148]). MSec, also called TNFAIP2 and B94, plays key roles in vital physiological processes (focal adhesion formation, vasculogenesis, inflammation, wound healing) was also established as a key component in the machinery required for TNT formation [149]. M-Sec interacts with the small GTPase RalA and serves as a key factor for TNT formation and function, particularly in macrophages [150,151,152]. M-Sec was described to promote nanotube formation by binding to RalA, which in turn interacted with the exocyst complex [150]. The RalA–RalBP1–Cdc42 pathway may play a role in the elongation of nanotubes, but is not central to their formation. The formation of long membrane protrusions like nanotubes would also require directed plasma membrane deformation, a process that could be mediated by myosin. Nanotubes allow cytoplasmic continuity between distant cells, a process which requires local membrane fusion. Leukocyte specific transcript 1 (LST1) acts as a membrane scaffold for the assembly of a multiprotein complex that orchestrates the formation of nanotubes [153]. LST1 recruits the small GTPase RalA and its effector filamin to the plasma membrane, and promotes the interaction between RalA and the exocyst complex. M-Sec myosin and myoferlin are additional LST1-interacting proteins that may contribute to the formation of open-ended nanotubes by locally deforming the plasma membrane and enabling membrane fusion. M-Sec also interacts with the ER chaperone ERp29. The chaperone activity of ERp29 was required for maintaining M-Sec protein stability [154]. Moreover, underlying mechanism of the TNT stability is the stability of phospholipid nanotubes [155].

Other mechanisms have been reported for TNT formation in different cell types. Thus, TNT biogenesis is not completely understood and may involve several protein complexes and signaling pathways that may differ according to the cell, probably reflecting a degree of cell-type specificity. Other alternative proteins and signal pathways regulating TNT formation in different types of cells have been reported, such as p53/Akt/PI3K/mTOR [156], Myosin10 [157], CDC42/IRSp53/VASP [158], and Rab11a/Rab8a [159]. Therefore, additional studies are needed to identify specific TNT regulatory factors depending on the cell type being studied in particular in ECs.

### 5.3. Induction of TNTs and Maintenance

There is a strong interest in the identification of the factors and conditions that enhance TNT formation and maintenance (Figure 2). Hypoxia, a condition that is strongly associated with tumor microenvironment and invasiveness, induces TNTs in various cancer cells [160]. While hypoxia promotes the formation of TNTs, TNTs in turn stimulate a potential positive feedback loop by mediating intercellular transfer of hypoxia-inducible factor-1α (HIF-1α) and vascular endothelial growth factor (VEGF) between connected cells, from cancer cells to ECs, to stimulate angiogenesis [147]. Thrombin induce TNTs in ECs [161]. Since cancer provides a prothrombotic state, this further supports a role for TNTs interplay between cancer cells and ECs. Cell-derived exosomes stimulated the formation of TNTs in cancer [147] and other cell types [162,163] and exosomes trafficking within and/or along TNTs to connecting cells has been visualized using time-lapse fluorescence microscopy [164]. Viruses from different families have been described to trigger formation of TNTs [133]

### 5.4. Roles of TNTs in Intercellular Communication and Transport

Our knowledge on the role of TNTs in ECs-to-cell communication is growing. TNTs permit intercellular communication between capillary ECs and myocytes in developing heart and skeletal muscle in vivo [165]. There is increasing evidence that the intercellular communication mediated by TNT is also involved in several pathologies, including cancer [166], neurodegenerative diseases [167], and infectious diseases [146,168]. TNTs may contribute to intercellular spread of pathogens, mostly viruses, but also bacteria that induce TNT formation in infected cells.

#### 5.4.1. TNTs and Cancer

TNTs are important for directional tumor cell streaming towards the endothelium. Previous studies have shown that while tumor cells close to the endothelium can migrate directionally, cells that are >500 μm away require the presence of macrophages to sustain their directional migration [169]. Using an elegant model that mimics cell streaming in vivo from the primary tumor towards the blood vessel, TNTs between macrophages and tumor cells are required for tumor cell streaming beyond a distance of 500 μm to directionally migrate towards the endothelium [170].

Metastatic cancer cells preferentially form nanoscale intercellular membrane bridges with ECs. The cancer cells use TNTs to horizontally transfer miRNA to the endothelium altering the EC phenotype through the regulation of the immunoregulatory receptors CD137 (4-1BB) and CD279 (B7-H 3) that feature tumor vasculature [171]. The ability to form the TNT with ECs correlates with the metastatic potential of the cancer cells. The use of pharmacological inhibitors to prevent TNT formation allows a reduction in the metastatic burden in experimental metastasis models [172], which suggests that targeting TNTs may be a promising approach in the management of metastatic cancer.

#### 5.4.2. TNTs and Viral Infections

The knowledge of TNT function has been improved by studies demonstrating the involvement of TNTs in virus intercellular spread. For some viruses, including retroviruses and alphaherpesviruses, the viral proteins involved in TNT formation have been identified [133]. For instance, HIV is able to trigger the formation of TNTs via the Nef protein. Some members of the alphavirus family, including Sindbis virus, Semliki Forest virus, and Chikungunya virus, induce actin- and tubulin-containing TNTs [168]. TNTs induced by alphaviruses depend on expression of viral structural proteins and were observed upon infection of primary HUVECs. Retroviruses, including HIV [146], but also Influenza viruses [173,174] and alphaherpesviruses [133], were shown to use TNTs for intercellular spread. Functionally, TNTs allow antibody-resistant infection of connected cells and also infection of a target cell which do not express the virus receptor thus extending cell tropism. However, spread of infection requires fusion-competent virus, indicating that virus transfer may not occur via direct cytoplasmic connectivity between donor and recipient cell [168].

#### 5.4.3. TNTs and Cargoes

TNTs have been shown to support cell-to-cell transfers of plasma membrane components, cytosolic molecules and organelles within cells. The presence of lipid droplets in the TNTs has been demonstrated in ECs indicating that lipid droplets represent another type of TNT cargo [175]. Lipid droplets are actively moving along TNTs in both directions. Lipid droplets can serve for the synthesis and the storage of lipids [176]. The number of lipid droplets increase in pathological conditions such as inflammation, cancer and hypoxia. In ECs, the biogenesis of lipid droplets increases under hypoxic conditions [177], a condition also associated with angiogenesis in tumors. Under angiogenic conditions (VEGF treatment), the number of lipid droplets increased significantly in microvascular ECs, while arachidonic acid not only increased the number of lipid droplets, but also tripled the extent of TNT formation [175]. A major role of lipid droplets in ECs is the synthesis of signaling lipids, such as eicosanoids (e.g., prostaglandins, leukotrienes and lipoxins), which are synthesized from arachidonic acid and regulate various cellular functions, such as inflammation, metabolism, cell activation, migration and apoptosis [178]. Direct intercellular transfer of lipid droplets may facilitate or propagate signaling by transferring the whole machinery for eicosanoid synthesis to the recipient cells.

As it was established that functional mitochondria can be transferred via TNTs between various cell types, studies have proven that this transfer may serve as a potent rescue mechanism to compensate for severe stress conditions (Figure 2). TNTs have been viewed as ‘emergency highways’ for the transport of vital organelles during situations posing a risk of apoptosis in damaged cells [179]. This phenomenon allows damaged/transformed/stressed cells to survive to cell death as shown in cancer cells [180]. Pasquier and colleagues demonstrated that TNTs allowed exchange of mitochondria from ECs to cancer cells inducing phenotypic changes and chemoresistance to cancer cells [181]. Mitochondrial transfer from bone marrow mesenchymal stem cells to HUVECs via TNTs as also been reported [182] and rescued the injured HUVECs by reducing apoptosis, promoting proliferation and restoring the transmembrane migration ability as well as the capillary angiogenic capacity of HUVECs. TNTs also display lysosomal transfer between distant cells. Yasuda and colleagues demonstrated that TNT formation allows to transfer intact lysosomes from EPC to apoptotic ECs, which resulted in rescue of ECs from premature senescence and apoptosis [183]. Lysosomal transfer is associated with the preservation of lysosomal pH gradient, functionally reconstituting lysosomal pool of stressed cells and improving endothelial cell viability. These authors further provided in vivo evidences of the role played by TNT formation between EPC and stressed vascular endothelium in improving endothelial dysfunction. These findings raise several important issues. EPC-mediated TNT formation reduces the senescence of ECs. This study also revealed that TNT-mediated exchange between EPC and ECs is selective toward damaged ECs and that repeated rounds of exchange occur between EPC and a number of ECs during a defined time period. Beyond the rescue of premature senescence by progenitor cells [183], heterotypic endothelial TNTs have been suggested to mediate, the rescue of injured ECs by stem cells [167,184], chemoresistance to mesenchymal cells [181] and the transfer of miRNA from smooth muscle cells [185].

#### 5.4.4. Homotypic Endothelial-To-Endothelial TNTs

TNTs can also occur between ECs (Figure 3), but we still have little understanding of the role of homotypic endothelial-to-endothelial TNTs or how they are formed. It has been suggested that ECs maintain Ca^2+^ signals during an injury on monolayer integrity by forming endothelial-to-endothelial TNTs [161] This induction depends on the highly sialylated CD31 protein, an adhesion molecule that is largely restricted to ECs. In cultured ECs (HUVEC) thrombin-that disrupt endothelial cell barrier and promote cell signaling and reorganization of the actin cytoskeleton readily causes the appearance of TNTs [161]. Together these findings suggest that endothelial-to-endothelial TNTs may involve the modulation of sialylated CD31 and the action of thrombin via protein kinase C (PKC)α.

#### 5.4.5. TNTs and Immune Responses

In addition to transport function, TNTs can mediate receptor–ligand interactions, particularly between immune cells [138]. The importance of TNTs in immune cell function and coordination during immune responses has been recently reviewed [186,187]. TNTs are able to connect different types of immune cells including B cells [188], T cells [146], macrophages [189], mast cells [190], NK cells [138] and dendritic cells [140]. TNTs can mediate transfer of MHC class I molecules between distant cells [191]. Chauveau et al. demonstrated that after 45 min of coincubation, 2–5% of primary human NK cells were connected to other primary NK cells, or various target cell lines (P815, 721.221 EBV-transformed B cells, and THP-1) [138]. In their model, NK cell TNTs were formed as cells disengaged after an initial close contact and thus may serve to sustain intercellular contacts over long distances rather than creating novel connections between cells. Engagement of the NK activating receptor NKG2D on NK cells with a ligand, the MCH class I-like molecules MICA, on target cells augment TNT formation. MICA also accumulates in TNT most probably to convey NK cytotoxocity. ECs express basal levels of NKG2D ligands including MICA and ULBPs and are upregulated upon stress and inflammation [131].

#### 5.4.6. TNTs as Therapeutic Targets

Considering the role of TNTs in intercellular communication, the use of TNTs as potential target to treat tissue injury, cancer, and infection is emerging (reviewed in [144]). Blocking TNT has been proposed to prevent cell-to-cell virus spreading and to control tumor chemoresistance and growth. Preliminary results suggested that inhibition of TNT formation could efficiently prevent the propagation of HIV viral particles [192] while inhibition of the TNT-mediated transfer of mitochondria leads to improved tumor cell death [193]. In another way, promoting TNT formation could provide an approach to protect cells from apoptosis, injury and senescence [183]. Another possible application for TNTs in clinics is their use for intercellular drug delivery. The delivery of doxorubicin, a chemotherapeutic drug, via TNTs has been achieved in vitro in various cancer cells [194]. However, both the efficacy and the specificity of delivery is questionable, and their improvement will require a better knowledge of TNT formation. Although promising, the development of effective TNT-interfering drugs, to inhibit or to induce TNTs, with minimal side effects and cytotoxicity is still challenging. Finally, the ability to use of TNTs for new therapeutic approaches will also rely on both the identification of TNT specific markers and the development of advanced methodologies to measure the efficacy of TNT-targeting drugs [144].

## 6. Conclusions

ECs possess a multilevel machinery to deliver messages to distant cells such as cells circulating in the blood vessel including platelets, leucocytes, other vascular cells, or normal and tumor cells in tissues. This intercellular crosstalk operates through the extracellular space and is orchestrated in part by the secretory pathway and the exocytosis of WPBs, secretory granules and EVs. WPBs and secretory granules allow both immediate release and regulated exocytosis of mediators implicated in the gatekeeper action of EC toward vascular homeostasis and permeability thrombosis, angiogenesis, inflammation, infection and immunity. The ectodomain shedding of transmembrane protein further provides the release of both receptor and ligands with key regulatory activities on target cells. EVs and in particular exosomes and TNT have emerged as an extensively studied mechanism of horizontal intercellular transfer of information [195]. EVs as well as most of molecules released by ECs might be useful, non-invasive biomarkers for the monitoring of EC and vascular function. Since endothelial EVs can efficiently deliver their cargo into target cells, they might be used as potential therapeutic agents to treat various diseases and injuries. Nevertheless, clinical applications will require further investigations to define with accuracy the uptake and targeting mechanisms as well as the content and functions of EVs in various contexts. Moreover, since ECs display specific phenotypes and functions that may vary according to the vascular bed and tissue [5,8], the impact of EC heterogeneity on secretome output, but also on TNTs, might require specific investigations.

Secreted mediators mediate a random pattern of intercellular communication that may not be predicted. In contrast, the communication via TNTs is more deterministic, by connecting a donor cell to a specific recipient cell. EVs and TNTs convey their messages via a large variety of cargoes, but also by direct receptor/ligand interactions initiating intracellular signaling in recipient cells or by external transport or molecular surfing on their surface. The potential role of TNTs on drug delivery has been proposed for cancer therapy [196]. Alternatively, the inhibition of TNT formation might be another issue to impair TNT-mediated chemoresistance in tumors. Consequently, a deeper understanding of the complex role of the secretome and TNT in the communication from and with ECs in specific vascular beds and tissues in the next years is warranted, allowing the development of new clinical applications.

## Figures and Tables

**Figure 1 ijms-22-07971-f001:**
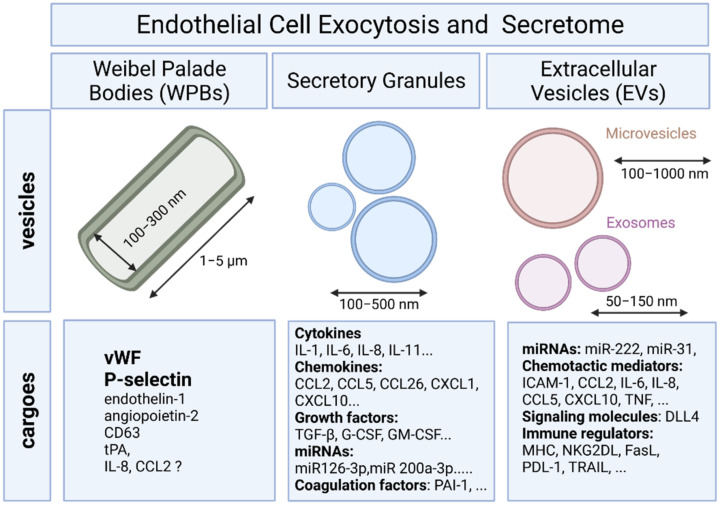
A schematic overview of secretory granules and extracellular vesicles and their cargoes in endothelial cells.

**Figure 2 ijms-22-07971-f002:**
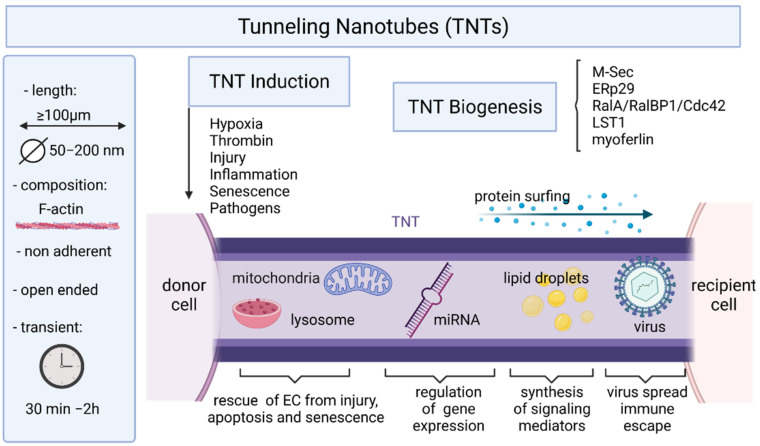
A schematic overview of TNT major characteristics and cargoes. Conditions inducing TNT formation and molecules implicated in TNT biogenesis are indicated.

**Figure 3 ijms-22-07971-f003:**
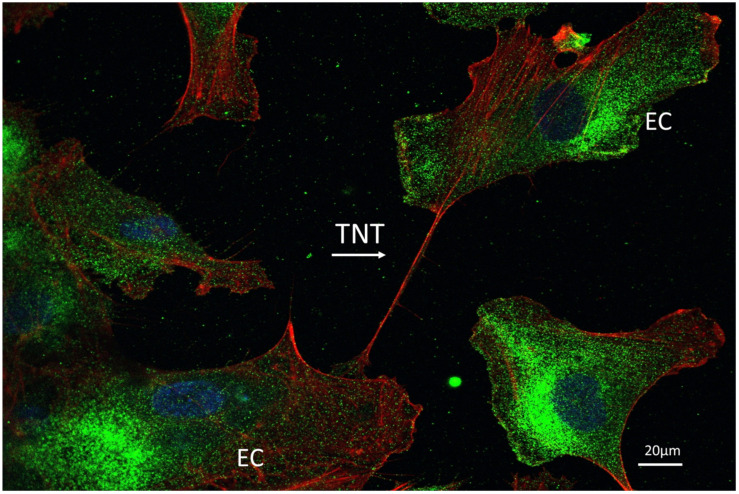
A representative confocal imaging of TNT between two endothelial cells. CoImmunostaining for F-actin (red) and MHC class I molecules (green) on human aortic EC culture. Nuclei are staining with DAPI (blue) and costainings are visualized as superimposition of images (magnification: 600×).

## Data Availability

Not applicable.

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
