# Peer review of "Secretome and Tunneling Nanotubes: A Multilevel Network for Long Range Intercellular Communication between Endothelial Cells and Distant Cells"

_ijms, 2021, doi:10.3390/ijms22157971_

Round 1
Reviewer 1 Report
This review by Béatrice Charreau titled “Secretome and Tunneling Nanotubes: A multilevel Network for Long Range Intercellular Communication between Endothelial Cells and Distant Cells” nicely summarizes the recent knowledge in extracellular vesicles formations and function with a focus on ECs. It also explores the Tunneling formations for long-distance communications between cells.
I found the review well-structured and organized, the figures nicely summarized the knowledge and help the reader. Overall recommend this review for publication with some minor comments:
Regarding TNTs a recent review published by the same journal describes in detail the TNT formation and function in disease (Han, X.;Wang, X. Opportunities and Challenges in
Tunneling Nanotubes Research: How Far from Clinical Application?. Int. J. Mol. Sci. 2021, 22, 2306. https:// doi.org/10.3390/ijms22052306).
This review should be at least cited and addressed.
Section 5.4 regarding TNTs, should probably be divided into subsections such as “TNTs in cancer” and “TNTs in infection and inflammation” with a final remark regarding possible translation in clinics (keeping in mind my previous comment).
There are some typos along with the manuscript and formatting problems with references:
See lines 407, and 411.
457 remove 1 during
457 “and”
535 induces
Author Response
Dear Reviewer,
Please find below my answers to your comments:
Regarding TNTs a recent review published by the same journal describes in detail the TNT formation and function in disease (Han, X.;Wang, X. Opportunities and Challenges in
Tunneling Nanotubes Research: How Far from Clinical Application?. Int. J. Mol. Sci. 2021, 22, 2306. https:// doi.org/10.3390/ijms22052306).
This review should be at least cited and addressed.
Re : Many thanks for this suggestion. This important and recent reference has been introduced in the revised manuscript.
Section 5.4 regarding TNTs, should probably be divided into subsections such as “TNTs in cancer” and “TNTs in infection and inflammation” with a final remark regarding possible translation in clinics (keeping in mind my previous comment).
Re : The 5.4 section has been modified as you suggested to include subsections and a final discussion on the possible translation in clinics.
There are some typos along with the manuscript and formatting problems with references:
See lines 407, and 411.
457 remove 1 during
457 “and”
535 induces
Re : Many thanks for these comments, these typos and formatting errors have been corrected as suggested.
Reviewer 2 Report
The review presented gives an overview on how endothelial cells communicate with distant cells through a multilevel network made by both secretome and tunneling nanotubes, organized together in a multilevel network. Due to the high number of information cited, reading is a little dispersive. Below are my observations and some suggestions:
- Figure 1. This figure should be enriched with some additional information: - target cells that preferentially receive biological information from the three different types of vesicles presented ( WPBS, Secretory Granules, EVs); - biological functions mainly related to each type of vesicles; - condition inducing vesicles formation and their biogenesis. Indeed, even if these concepts are written in the text, they are not summarized at all.
- Paragraph 3.4. (326-329) please rephrase this sentence, it is not clear
- Both pharagrafs “ EVs trigger endothelial protection and vascular repair” and “Transfer of miRNAs via EVs” are numbered 3.4
- Paragraph 3.5 “EVs and immune response” is not informative on the cross-talk between endothelial and immune cells through vesicles or nanotubes. It may be appropriate to discuss the information written in the paragraph “Transfer of miRNA via EVs” (from line382 to
- 395) in this section on immune responses. Furthermore, original and recent papers about exosomes containing molecules with specialized function in immune response should be cited (lines 398-403). Author highlighted that EVs are investigated as regulator of immune response mainly in cancer, so it should be appropriate to contextualize that the crosstalk between ECs and Immune cells is studied mainly in cardiovascular disease and infections. Please highlight recent works on this (i.e. “PMID:34150006; PMID: 34128973; PMID: 31076589
- Can you please add images of EVs (i.e TEM) as you made for nanotubes? (fig 3)
Author Response
Dear Reviewer,
Please find below my answers to your comments:
The review presented gives an overview on how endothelial cells communicate with distant cells through a multilevel network made by both secretome and tunneling nanotubes, organized together in a multilevel network. Due to the high number of information cited, reading is a little dispersive. Below are my observations and some suggestions:
- Figure 1. This figure should be enriched with some additional information: - target cells that preferentially receive biological information from the three different types of vesicles presented ( WPBS, Secretory Granules, EVs); - biological functions mainly related to each type of vesicles; - condition inducing vesicles formation and their biogenesis. Indeed, even if these concepts are written in the text, they are not summarized at all.
- Re : I acknowledge that the synthetic view provided in the figure 1 is partial but I believe that adding more informations could be confusing.
- Paragraph 3.4. (326-329) please rephrase this sentence, it is not clear
- Re : This point has been corrected.
- Both pharagrafs “ EVs trigger endothelial protection and vascular repair” and “Transfer of miRNAs via EVs” are numbered 3.4
- Re : This point has been corrected.
- Paragraph 3.5 “EVs and immune response” is not informative on the cross-talk between endothelial and immune cells through vesicles or nanotubes. It may be appropriate to discuss the information written in the paragraph “Transfer of miRNA via EVs” (from line382 to395) in this section on immune responses. Furthermore, original and recent papers about exosomes containing molecules with specialized function in immune response should be cited (lines 398-403). Author highlighted that EVs are investigated as regulator of immune response mainly in cancer, so it should be appropriate to contextualize that the crosstalk between ECs and Immune cells is studied mainly in cardiovascular disease and infections. Please highlight recent works on this (i.e. “PMID:34150006; PMID: 34128973; PMID: 31076589
Re : Many thanks for these interesting comments and suggestions. As suggested this paragraph has been modified and now includes more investigations and the related references (page 10).
- Can you please add images of EVs (i.e TEM) as you made for nanotubes? (fig 3)
- Re : Unfortunately, I do not currently work on EVs and consequently I have no image available.
Reviewer 3 Report
The author has presented a thorough review of accumulated knowledge on cell-derived nanostructures that mediate intercellular communication involving endothelial cells. The review also includes consideration of mechanisms that are of general importance in intercellular communication. The role of Weibel Palade Bodies, secretory granules, extracellular vesicles and tunelling nanotubules is elucidated. I suggest that the authors consider the comments below:
In considering extracellular vesicles (EVs), the author states the descriptions that have been repeated in many papers, i.e. that there are three types of EVs: apoptotic bodies, microvesicles and exosomes; microvesicles are formed by the budding of the plasma membrane while exosomes are formed in internal cell compartments and are released into the cell exterior by fusion of these compartments with the plasma membrane. While these are hypothetical mechanisms, in actual samples that are harvested from different biological sources, a population of small particles can be found that is heterogeneous in size, shape and composition and when already present in the isolate, the origin of the particles is usually not evident. Formation of EVs due to processing of samples is often overlooked although it may present a considerably portion of EVs in the samples.
Page 271, Line 227: The author states: "Endothelial EVs account for approximately 5–15%, and constitute a large subclass of all circulating EVs in blood, albeit the majority of circulating plasma EVs are derived from platelets and erythrocytes". Please cite these statements. Have these claims been supported by measurements directly in circulation? In many works, the authors tacitly assume that what they measure in blood samples that are taken out of the body are the circulating particles. Also, it is often assumed that binding of marker molecules to the particles exclusively identifies the origin of the particles. As these generalizations are being repeated many times by many authors, they are becoming a dogma, which prevents development of the model approach to the description of small particles shed from cells and the underlying mechanisms.
To be more precise, the author could rephrase her claims to something like: "The levels of EVs carying receptors for binding XXX have been found elevated...(citation)"
Page 271, Line 239: The authors state: "MVs are large vesicles (100-1000 nm) that may display a diverse range of sizes and are released from the plasma membrane by budding. In contrast, exosomes are smaller vesicles (< 150 nm) which originate from the endosome [57]." If the two populations are divided solely by size, where do the particles sized between 100 and 150 nm fit? Taking a particular particle in the isolate sized say 120 nm, what could determine whether it is a microvesicle or an exosome? Then, isolates from many different systems yield particles of such sizes.
Page 272, Line 1: Then, however, the author states: "The mechanisms of EV uptake and cargo delivery into the cytosol of recipient cells remain incompletely elucidated." As it is written, this indicates that things are quite clear about EV classification, formation mechanisms and contents while the cargo delivery mechanisms are less clear. Please reconsider whether this is really so; I argue that also the EV formation mechanisms are rather poorly understood.
I suggest that the auhor critically reconsiders the state of art in EV knowledge and replaces the claims and conclusions that seem too bold by expressions such as "It was suggested that ....", "It was found that .....", It was speculated that ...". I think that such point of view would bring benefit to the field of EVs in breakthrough to the clinical and industrial use. This is being promissed already for years but did not yet take place, one of the reasons being poor understanding of the processes and therefore poor repeatability of the procedures. Furthermore, the distinction between the measurements made on samples and interpretations would be of additional help. Please have in mind that the samples are being importantly modified by the harvesting and assessment procedures. Mentioning this in the Discussion would be of help to the readers.
The author states: "The role of TNTs in ECs-to-cell communication started to be elucidated in the past decade." I suggest to rephrase this as these features have been considered also before (evidenced also in the references given by the author and in the therein contained references).
The underlying mechanism of the TNT stability is the stability of phospholipid nanotubes which could be mentioned in the discussion. Please see Drab M., Stopar D., Kralj-Iglič V., Iglič A. Cells. 8, 6: 1-17. 2019.
Author Response
Dear Reviewer,
Please find below my answers to your comments:
In considering extracellular vesicles (EVs), the author states the descriptions that have been repeated in many papers, i.e. that there are three types of EVs: apoptotic bodies, microvesicles and exosomes; microvesicles are formed by the budding of the plasma membrane while exosomes are formed in internal cell compartments and are released into the cell exterior by fusion of these compartments with the plasma membrane. While these are hypothetical mechanisms, in actual samples that are harvested from different biological sources, a population of small particles can be found that is heterogeneous in size, shape and composition and when already present in the isolate, the origin of the particles is usually not evident. Formation of EVs due to processing of samples is often overlooked although it may present a considerably portion of EVs in the samples.
Re : Many thanks for this pertinent comment. As suggested, this comment has been introduced in the revised version (page 6).
Page 271, Line 227: The author states: "Endothelial EVs account for approximately 5–15%, and constitute a large subclass of all circulating EVs in blood, albeit the majority of circulating plasma EVs are derived from platelets and erythrocytes". Please cite these statements. Have these claims been supported by measurements directly in circulation? In many works, the authors tacitly assume that what they measure in blood samples that are taken out of the body are the circulating particles. Also, it is often assumed that binding of marker molecules to the particles exclusively identifies the origin of the particles. As these generalizations are being repeated many times by many authors, they are becoming a dogma, which prevents development of the model approach to the description of small particles shed from cells and the underlying mechanisms.
Re : The text has been changed page 6 to adress this concern and two references have been added (Diehl et al., , 2012 and Gardiner et al., 2016).
To be more precise, the author could rephrase her claims to something like: "The levels of EVs carying receptors for binding XXX have been found elevated...(citation)"
Page 271, Line 239: The authors state: "MVs are large vesicles (100-1000 nm) that may display a diverse range of sizes and are released from the plasma membrane by budding. In contrast, exosomes are smaller vesicles (< 150 nm) which originate from the endosome [57]." If the two populations are divided solely by size, where do the particles sized between 100 and 150 nm fit? Taking a particular particle in the isolate sized say 120 nm, what could determine whether it is a microvesicle or an exosome? Then, isolates from many different systems yield particles of such sizes.
Re : I acknowledge that cells release a large range of EVs which are heterogene in size. The size generally described for exosomes is 50–150 nm thus overlapping with the size of small MVs. Indeed, the size of EVs is a suffisant
Page 272, Line 1: Then, however, the author states: "The mechanisms of EV uptake and cargo delivery into the cytosol of recipient cells remain incompletely elucidated." As it is written, this indicates that things are quite clear about EV classification, formation mechanisms and contents while the cargo delivery mechanisms are less clear. Please reconsider whether this is really so; I argue that also the EV formation mechanisms are rather poorly understood.
Re ; Thanks for this comment, as suggested this sentence has been modulated (Pages 6 and 7).
I suggest that the auhor critically reconsiders the state of art in EV knowledge and replaces the claims and conclusions that seem too bold by expressions such as "It was suggested that ....", "It was found that .....", It was speculated that ...". I think that such point of view would bring benefit to the field of EVs in breakthrough to the clinical and industrial use. This is being promissed already for years but did not yet take place, one of the reasons being poor understanding of the processes and therefore poor repeatability of the procedures. Furthermore, the distinction between the measurements made on samples and interpretations would be of additional help. Please have in mind that the samples are being importantly modified by the harvesting and assessment procedures. Mentioning this in the Discussion would be of help to the readers.
Re : As suggested, attention has been paid to modulate or attenuate the claims (page 6 and 7).
The author states: "The role of TNTs in ECs-to-cell communication started to be elucidated in the past decade." I suggest to rephrase this as these features have been considered also before (evidenced also in the references given by the author and in the therein contained references).
Re : As suggested, this sentence has been rephrased (page 13).
The underlying mechanism of the TNT stability is the stability of phospholipid nanotubes which could be mentioned in the discussion. Please see Drab M., Stopar D., Kralj-Iglič V., Iglič A. Cells. 8, 6: 1-17. 2019.
Re : This comment and the reference have been added to the revised manuscript (page 12).